# Micronutrients and Breast Cancer Progression: A Systematic Review

**DOI:** 10.3390/nu12123613

**Published:** 2020-11-25

**Authors:** Olga Cuenca-Micó, Carmen Aceves

**Affiliations:** Instituto de Neurobiología UNAM-Juriquilla, Querétaro 76230, Mexico

**Keywords:** breast cancer, micronutrients, cancer

## Abstract

Epidemiological studies on micronutrient consumption have reported protective associations in the incidence and/or progression of various cancer types. Supplementation with some of these micronutrients has been analyzed, showing chemoprotection, low toxicity, antiproliferation, and the ability to modify epigenetic signatures in various cancer models. This review investigates the reported effects of micronutrient intake or supplementation in breast cancer progression. A PubMed search was conducted with the keywords “micronutrients breast cancer progression”, and the results were analyzed. The selected micronutrients were vitamins (C, D, and E), folic acid, metals (Cu, Fe, Se, and Zn), fatty acids, polyphenols, and iodine. The majority of in vitro models showed antiproliferative, cell-cycle arrest, and antimetastatic effects for almost all the micronutrients analyzed, but these effects do not reflect animal or human studies. Only one clinical trial with vitamin D and one pilot study with molecular iodine showed favorable overall survival and disease-free interval.

## 1. Introduction

Since the 1980s, there has been increasing epidemiological evidence of the relationship between inadequate intake of micronutrients and the appearance of tumor processes [1]. However, the heterogeneity of compounds, models, and types of malignancy makes it impossible to assign general effects. Breast cancer is the most common cancer in women, affecting more than two million each year. In 2018, more than 600,000 women died from breast cancer, representing approximately 15% of all cancer deaths among women worldwide [2]. There is extensive literature on nutrition and the risk of breast cancer [3,4,5,6]; however, because some nutrients and hormones play a dual role in tumor initiation and progression, these processes need to be analyzed separately [7,8]. In this review, we focus on the effects of micronutrients once the breast tumor process has been established. Given our group’s interest in molecular iodine (I_2_), we compared the effects and mechanisms proposed for supplementing this halogen with those of the other micronutrients. All nutrients have antitumor properties in cellular cultures; however, when scaling to in vivo models, most of the micronutrients lose these properties. Only vitamin D (Vit D) and I_2_ supplements showed associations with improved overall survival (OS) and disease-free interval (DFI) in clinical trials [9,10].

Regarding the mechanisms, most of the micronutrients analyzed in this review work as antioxidants, reducing the aberrant redox environment characteristic of tumor processes. Other actions have also been described in some components. Vit D acts as a genomic regulator and folates are involved in purine/pyrimidine synthesis and methylation reactions. I_2_ appears to act as a genetic modulator by joining lipids and activating nuclear receptors. Its ability to modify DNA methylation processes has also been proposed. In the progression of breast cancer, I_2_ seems to be the best micronutrient for adjuvant therapy with different antitumor mechanisms, but more extensive clinical trials are needed.

## 2. Materials and Methods 

A search of the scientific literature was carried out in PubMed on August 28, 2019, with the terms “micronutrients breast cancer progression”, which yielded 425 results. All 425 papers were screened with the following inclusion criteria: original articles written in English, analyzing one or more micronutrient supplement effects in breast cancer progression. A total of 352 articles were excluded: articles published in languages other than English (15 articles), reviews (65), papers about synthetic micronutrient analogs or without micronutrients (204), articles concerning other cancers (39), articles in which there was no cancer progression (15), and methodology-aimed articles (14). Finally, 76 articles were assessed for eligibility, and 73 were included in this review. Metabolites such as carotenoids, Vit A, or coenzyme Q10 were less represented in the articles and are not analyzed. 

## 3. Results

If we consider the percentage of papers on each nutrient, 37% were about Vit D or its natural metabolites, 27% were about metals such as copper (Cu), iron, selenium (Se), and zinc (Zn), 9% and 7% were about folates and vitamin C (Vit C), respectively, and 5% of articles analyzed in this review were on polyphenols, fatty acids, vitamin E (Vit E), and iodine. These data provide an overview of current research interest in nutrient supplementation and breast cancer progression. Next, the main results of our analysis are detailed.

### 3.1. Vitamin D

Vit D is characterized as a vitamin (i.e., a compound with the catalytic activity of biological processes); however, due to its metabolism and action mechanisms, it is considered as a pro-hormone. Reviews of Vit D’s physiological actions suggest that its active form, calcitriol, regulates calcium and phosphate homeostasis and plays a key role in the physiology of various organs and systems. Calcitriol was mostly studied in the immune system as an immunomodulator targeting various immune cells, including monocytes, macrophages, dendritic cells, T-lymphocytes, and B-lymphocytes (B) [11,12]. Calcitriol’s mechanisms of action start when it binds to the Vit D receptor (VDR). This nuclear receptor can form homodimers (VDR–VDR) or heterodimers (with the X retinoid receptor: CDR–RXR) that bind to specific genome responsive elements (VD-REs or VD/RXREs) to activate the transcription of their target genes [13]. Vit D is not the only micronutrient capable of activating these transcription factors; low-affinity nutritional ligands for VDR, such as curcumin, unsaturated fatty acids, and anthocyanins, have been described with the ability to activate this route. Others, such as resveratrol, have been described as VDR signaling enhancers [14]. The activity of VDR can be modulated epigenetically by histone acetylation. It cooperates with other nuclear receptors that are influenced by histone acetyltransferases (HATs) and several types of histone deacetylases (HDACs). HDAC inhibitors (HDACi) and demethylating drugs may contribute to Vit D metabolism [15]. 

Concerning cancer, studies (most of them in vitro) demonstrated antineoplastic effects of calcitriol. The molecular mechanisms include inhibition of thekinase-interacting serine/threonine-protein kinases MAPK–ERK pathway, suppression of Epidermal Growth Factor (EGFR) and Insulin-like growth factor 1 (IGF1), and inhibition of telomerase, Bcl-2, and Myc expression [16]. Specifically, in breast cancer, the maximum dose of supplementation was 100 nM, which seems sufficient to inhibit cell proliferation, increase apoptosis markers, and modify intracellular glucose metabolism parameters [17,18,19,20]. These studies showed, regardless of the cell type used, antiproliferative effects, increased redox potential, and cell-cycle arrest in G1. Most of these studies were performed on MCF-7 cells (estrogen receptor-positive: ER^+^), and the supplement of this micronutrient was accompanied by significant decreases in the expression of the aromatase enzyme, estrogen receptor (ER), cyclooxygenase type 2 (Cox-2), prostaglandin E2 (PGE2), and the antiapoptotic protein Bcl-2. These prostaglandins are also associated with increases in the number of apoptotic cells [20,21,22,23,24,25,26]. Furthermore, studies that used breast cell lines of normal origin with the mutated Ras oncogene (MCF10A-ras) found a pro-oxidant effect (via inhibition of the enzyme Pyr carboxylase) and a decrease in the flow of glutamine (Gln) inside the cell, generating a significant decrease in proliferation [27,28,29]. In general, calcitriol restores reactive oxygen species (ROS) equilibrium in tumor cells, reduces cancer proliferation, and has a relevant role in apoptosis and autophagy [30,31,32]. In preclinical studies, the findings are more complex and dependent on breast cancer type and doses. The differentiated cancer model (estrogen+), with a moderate concentration of Vit D supplement, did not exhibit any effects [25,28], exhibited inhibition of tumoral growth accompanied by decreases in the expression of Bcl-2, aromatase, estradiol, and Cox-2 [20], or presented an increase in apoptosis cell content with augmented p53 expression [21]. In contrast, in the triple-negative breast cancer model (immunosuppressed mice AT1^+^ cells), Vit D supplementation increased the tumors’ metastatic potential [33]. Finally, studies in cancer patients described that high doses (250 µg/day) did not affect tumor progression, while low supplementation (10 µg/day) reduced mortality by at least 20% [9,34]. It also has been described that Vit D deficiency or the inability to synthesize its active compound (calcitriol) accelerates tumor growth [35,36]. However, some studies showed that Vit D supplementation to restore plasma levels (time and doses) stops the tumoral growth [34], whereas, in other studies using high concentrations (1250 µg/week) sufficient to revert these deficiencies, it could not decrease cancer progression [37]. In general, calcitriol alters ROS equilibrium in cancer, reduces cancer proliferation, and has a relevant role in apoptosis and autophagy [38,39,40]. It appears that high serum levels are associated with better survival and DFI [41,42,43,44,45]. In 2010, the Institute of Medicine of the US National Academy of Sciences recommended an intake of 600 IU/day of Vit D (15 µg) to maintain adequate serum 25(OH)D for normal bone mineralization. However, the recommendation does not include the extra-skeletal effects of Vit D. On the other hand, the Endocrine Society committee suggests higher doses (1000–2000 UI equivalent to 25–50 µg) to correct deficiencies and prevent fractures and does not consider this supplementation to prevent cancer or cardiovascular diseases [46]. These descriptions agree with Goulao et al.’s recent review, which included 30 clinical studies with more than 18,000 participants, finding no evidence that Vit D supplementation alone reduces the incidence of cancer or mortality in established cancer processes [47].

### 3.2. Metals

#### 3.2.1. Copper

Cu is a transition metal that the body requires as a catalytic cofactor or a structural protein component involved in redox reactions. It participates in the adequate synthesis of some metabolites, such as hemoglobin, elastin, and collagen, as well as in transporting oxygen to the mitochondrial respiratory chain. The immune system also requires Cu to perform several functions; animal models and cell cultures have been used to assess the role of Cu in the immune response. Some research showed that Cu deficiency is accompanied by a reduction in T-cell proliferation and interleukin production [48]. Lowering Cu levels in the diet increases protein oxidation and DNA methylation [49]. There are numerous Cu-dependent enzymes whose activity diminished with Cu deficiency: ceruloplasmin, superoxide dismutase (SOD), cytochrome C oxidase (COX), and ascorbic acid oxidase, among others [50]. At the molecular pathway level, Cu directly affects the phosphorylation of extracellular signal-regulated kinase (Erk) by meiotic chromosome axis-associated kinase 1 (Mek1). Mek1, a MAPK pathway kinase, has two binding sites for Cu, and its presence increases Erk phosphorylation in a dose-dependent manner [51]. For this micronutrient, the recommended daily intake is 900 µg, while the maximum tolerable level is 10 mg/day [52]. Many types of cancer have elevated intracellular Cu levels or exhibit alterations in this metal’s systemic distribution [53]. In breast cancer, elevated serum Cu levels correlate with the stage and progression of the disease [54], and tumor cells have four times more Cu than healthy breast cells [55]. The addition of Cu to breast cancer cells showed opposite results depending on the cell type. In triple-negative MDA-MB-231 cells with mutated p53, the Cu supplement increased proliferation and survival and Akt phosphorylation [56].

In contrast, in MCF-7 cells positive for ER, this micronutrient increased p53 phosphorylation and the expression of p21, resulting in cell-cycle arrest in G1 and apoptosis [56]. Four articles on clinical studies exploring the relationship of Cu to breast cancer progression were analyzed for this review (see Appendix A
Appendix A). Serum and hair Cu levels were higher in patients than in healthy people [57,58]. Angiogenic and metastatic properties are attributed to this metal. In breast tumors, a Cu-dependent RedOx protein Memo has been reported to play an essential role in migration and metastasis (increasing intracellular ROS levels) [59]. Furthermore, a clinical study evaluating a Cu chelator effect showed a reduction in angiogenic markers [60]. 

#### 3.2.2. Iron

Iron, another transition metal, acts in mammals as a cofactor of hemoproteins (hemoglobin, catalase, peroxidases, cytochromes) involved in oxygen binding, transport, and metabolism. It is also a cofactor of other proteins (without a heme group) with functions in DNA synthesis, cell proliferation and differentiation (ribonucleotide reductase), gene regulation, drug metabolism, and steroid synthesis [61]. The peptide hormone hepcidin is the primary regulator of iron metabolism in the body. This enzyme increases its expression with high hepatic iron levels, with increased levels of iron in the plasma, and during inflammation (by Interleukin 6 (IL-6) in a mechanism that involves the activation of Signal transducer and activator of transcription 3 (STAT3) [62]. In contrast, its expression is suppressed under hypoxic conditions [63]. Overall, in tumors, elevated intracellular iron levels are reported compared to their healthy cellular counterpart. Excess iron favors tumor growth [64]; thus, depleting this metal, either by reducing dietary intake or by chelating, shows inhibition of tumor growth [65]. Specifically, epidemiological studies show positive associations between dietary iron consumption and breast cancer incidence [66]. The studies analyzed in this review (Appendix A) show results consistent with those reported in the literature. Both tumor tissue and trace element quantification showed higher iron levels than healthy tissue or patients [60,67]. At the mechanistic level, in silico models suggest oncogenic Ras pathways in altered iron homeostasis in tumors [68]. Studies with cell cultures show that the most aggressive lines accumulate a more significant amount of iron [69]. The use of chelators inhibits breast carcinoma growth and causes cell-cycle arrest in the S phase accompanied by apoptosis [70].

#### 3.2.3. Selenium

Se is a metalloid with both nutritional and toxic properties. In humans, Se’s nutritional functions are carried out through 25 enzyme proteins with selenocysteine in its active center [71]. These selenoproteins have a wide range of pleiotropic effects, from antioxidant (such as glutathione peroxidases) to anti-inflammatory effects (selenoproteins) to the activate/deactivate thyroid hormones (deiodinases) [72]. The epigenetic evidence indicates that high Se exposure leads to DNA methyltransferase expression/activity [73]. The daily intake recommendation is set at 60 µg for men and 53 µg for women [74]. In cancer, Vinceti et al. [75] analyzed in a recent review 55 prospective observational studies and concluded that there is a lower incidence and mortality associated with high exposure to Se. However, in an analysis of eight clinical trials, no clear evidence was found that supplementation with Se reduced the risk of any cancer [76]. In the present review, we found five articles on Se and breast cancer (Appendix A). Three of these articles were on prospective studies [57,70,77], and the serum of Se showed lower levels in all patients than in their healthy counterparts. On the contrary, in tumors, the Se levels were higher than in the adjacent tissues. Another finding in various studies is that the decreases in circulating levels of Se correlate with the stages of disease progression [75]. The analyzed preclinical studies showed an inverse relationship between circulating levels of Se and vascular epithelial growth factor (VEGF) [78]. Se supplementation seemed to inhibit tumor progression in preclinical studies and cell cultures [79]. A decrease in Se levels appears to be widespread in cancer progression, but there is no evidence of its benefits as an adjuvant in tumor progression.

#### 3.2.4. Zinc

Zn is an omnipresent trace element. It is found in all tissues of the body, where its most significant role is in stabilizing the structure of many proteins. This element has three main functions in the organism: catalytic (DNA synthesis, brain development, and wound healing), structural (DNA replication), and regulatory (enzymatic activity and protein stabilization). Among its many functions in the body, Zn is involved in immune response, oxidative stress, apoptosis, and aging [80]. At the regulatory level, Uciechowski and his group described epigenetic and redox-dependent mechanisms as responsible for Zn effects in the immune system [81]. Various studies established an association between Zn deficiency and cancer (in cell cultures, preclinical models, and human studies) [82,83]. In our review (Appendix A), two studies examined the amount of systemic Zn in patients with breast cancer. Measurements in serum and hair indicated a decrease in Zn compared to their healthy counterparts [57,84].

Regarding intratumor Zn levels, the results vary depending on the technique used for metal detection [85,86]. Our analysis showed that the overexpression of the Zn transporter ZIP10 in tumor cells [87] and the chelation of this metal inhibit the invasiveness of several metastatic tumor cell lines [56,87]. Like Se, systemic Zn levels decrease with the disease; however, high intratumoral concentrations occur, which is explained by the overexpression of its ZIP10 transporter. A therapeutic approach used in recent years involves Zn chelators because Zn is required for tumor cell adaptation to hypoxic conditions [88].

### 3.3. Folates

Folates are micronutrients of the vitamin B complex. They are acceptors/receptors of 1-carbon units and function as coenzymes involved in purine/pyrimidine synthesis and various methylation reactions [89,90]. Associations have been reported between the nutritional status of folate and chronic diseases such as cardiovascular disease, cancer, and cognitive dysfunction [91]. In cancer, depending on the time of supplementing with folate, the results could be the opposite. Thus, supplementation before the existence of preneoplastic lesions can prevent tumor development, whereas supplementation in the presence of established lesions increases tumorigenesis [92,93,94]. This dual role of folate in carcinogenesis has been explained as an adequate intake of folates prevents DNA damage [95], while excess folate during an established tumor process decreases the expression of tumor suppressor genes [96,97]. In this review (Appendix A), studies in preclinical models showed that high-folate diets increase tumor volume and the number of tumors, while deficiency of this nutrient significantly inhibits breast cancer [98,99,100]. The mechanisms via which these results have been explained are inferred from in vitro studies where folic acid supplementation increased the expression of enzymes responsible for DNA methylation and the decrease in tumor suppressor genes [Phosphatase and tensin homolog;(PTEN) and adenomatous polyposis coli (APC)] associated with increased methylation of its promoters [101]. However, studies in human patients showed conflicting results. Two studies analyzed folate consumption using a dietary questionnaire; one found an inverse association between folate consumption and mortality [102] and the other did not find any association [103]. Another study analyzing different subtypes of breast cancer found that plasma folate levels are lower in patients with human epidermal growth factor receptor 2 (HER2^+^) and triple-negative cancer [104]; however, more in-depth studies are needed in this area.

### 3.4. Vitamin C

Vit C (or ascorbic acid) is an essential micronutrient present in citrus and other vegetables. Its biological functions are extensive, as they contribute to the synthesis of metabolites (carnitine, catecholamine, norepinephrine, etc.) and collaborate in the metabolism of tyrosine, tryptophan, folic acid, and cholesterol. They also participate in collagen formation and maintenance and thyroid hormone (TH) synthesis [105,106,107,108]. On the other hand, ascorbic acid supplementation strengthens the immune system, increasing neutrophil motility, leukocyte transformation, and phagocytosis [108]. It is also a potent antioxidant [109]; this capacity makes it a supplier of reduced iron, necessary for epigenetic regulation of DNA and histone demethylation [110]. In cancer patients, Vit C deficiency is common. It has been reported that pharmacologic doses of ascorbate act as a pro-oxidant ascorbate radical, decreasing the growth and aggressiveness of ovarian, pancreatic, and glioblastoma xenografts in mice [111]. In the articles about this micronutrient in breast cancer (Appendix A), culture and preclinical models indicated that Vit C deficiency facilitates tumor growth and expansion, while supplementation reduces cell proliferation [112,113]. In studies in human patients, consumption before malignancy appeared to be associated with survival; however, once the tumor process was established, this protective effect seemed to disappear [114]. The progression of malignancy is related to decreased serum ascorbic acid, which is exacerbated if the patients have been smokers (exposed to a more significant amount of oxidants) [115,116]. A single clinical study evaluated the effect of intravenous injections of Vit C in breast cancer patients. The maximum dose they supplemented was 50 g, where plasma ascorbate concentrations averaged 18 mM. Under these conditions, they observed a reduction in serum inflammatory markers such as IL-1α, IL-2, IL-8, and tumor necrosis factor alpha (TNF-α) and a reduction in C-reactive protein levels associated with poor prognosis and worse survival rates [117]. Systemic Vit C levels are inversely related to exposure to oxidants (such as tobacco). This decrease is also observed during disease progression, which could be explained, in part, by the increase in ROS activity characteristic of tumor progression [115,116]. These effects, such as the redox pair of Vit C, depend on its concentration. An antioxidant effect is observed at a physiological range of serum Vit C between 26 and 84 µM (equivalent to an intake of 75–90 mg/day). To achieve an oxidizing effect, at least >100 µM in plasma is required, which is only achieved with intravenous injections of ascorbic acid (117). The results analyzed in this review point to a possible benefit of ascorbic acid supplementation, although clinical studies are needed to verify the effects observed in other models.

### 3.5. Polyphenols

Polyphenols are a group of natural compounds with phenolic structural characteristics. More than 8000 structures have been identified and are present in fruits and grains [118]. High consumption has been linked to a lower risk of cancer, cardiovascular diseases, chronic inflammation, and degenerative diseases [119,120]. The primary biological role of polyphenols is associated with their antioxidant properties; however, they have also been described as metal chelators (such as Fe^2+^), anti-inflammatories, and promoters of probiotic actions [121,122,123]. In cancer, the protective effect of polyphenols is debated due to the discrepancy between study models and the use of non-physiological concentrations [124]. Although numerous possible mechanisms have been elucidated, most of the results obtained show different effects at low or high supplement concentrations [125]. These biphasic effects could be explained by their ability to modulate hormonal receptors; the chemical structure of polyphenols defines their affinity for binding to ERs. This affinity is lower than that of estradiol and allows agonist or antagonist reactions depending on the bound polyphenol (e.g., genistein is an ERα and ERβ agonist; resveratrol is an ERα antagonist and ERβ agonist) [126,127,128]. All the studies analyzed in this review on the effects of polyphenols on breast cancer progression (Appendix A) were conducted in cell cultures and reflect the discordant results indicated in the literature. In the MCF-7 cell model (ER^+^), supplementation with small amounts of isoflavones showed increased cell proliferation [128]. At the same time, high doses inhibited growth in a dose-dependent manner and stopped the cell cycle in G1 [129]. In MDA-MB-231 cells (triple-negative model), small amounts of luteolin did not affect invasive and cell migration capabilities [130], while small quantities of naringin (0.1 µM) showed significant inhibitory competences [131]. The studies analyzed in this review do not offer a clear direction on the use of polyphenols to treat breast cancer, which is consistent with what has been published for other types of malignancies.

### 3.6. Fatty Acids

The fatty acids analyzed in this review are α-lipoic acid and α-linolenic acid. α-Lipoic acid is found in high concentrations in spinach, broccoli, liver, and kidney and participates in the energy metabolism of carbohydrates, proteins, and fats [132]. It is also a cofactor for mitochondrial enzymes and a potent antioxidant [133]. As a structural component of cell membranes, the location and organization of α-lipoic acid and α-linolenic acid within cellular lipids directly influence the behavior of several proteins involved in immune cell activation [134]; in fact, Jacobsen et al. associated a lower level of DNA methylation in inflammatory disease and inflammatory response with a high-fat diet [135]. The effects of dietary fatty acids have been described in numerous signaling pathways in tumorigenesis, inhibiting tumor growth and proliferation and inducing apoptosis [136]. a-Linolenic acid is present in walnuts, canola, many legumes, and green leafy vegetables [137]. It is a precursor of omega-3 fatty acids and is essential for brain development and functions, cardiovascular health, and inflammatory response [138,139]. In tumor cell cultures, antitumor properties are attributed to α-linolenic acid, due to the decrease in VEGF and metalloprotease expression and the restoration of tumor suppressor gene expression (e.g., Rb and p. 53) [140]. The same is observed with lipoic acid (Appendix A), which exhibited antitumor effects such as decreased ROS, cell-cycle arrest followed by apoptosis, and decreased proliferation [141,142]; however, when scaled in preclinical models, similar doses generated contradictory responses. In mice with HER2 overexpression, supplementation with lipoic acid increased tumor growth [143], whereas, in nude mice xenografts, the treatment significantly retarded tumor growth [144]. Very few studies were found on fatty acids in breast cancer progression and none in patients. The results of these studies show antitumor effects in cell lines; further studies in preclinical models are necessary to establish possible benefits.

### 3.7. Vitamin E

Vitamin E (Vit E) is a group of eight fat-soluble compounds: four tocopherols (α, β, γ, and δ) and four tocotrienols (α, β, γ, and δ). Tocopherols predominate in olive, sunflower, corn, and soybean oils, while tocotrienols are found in palm oil or rice bran [145,146]. These compounds exert antioxidant, neuroprotective, and cholesterol-lowering activities [147]. Vit E is found in higher concentrations in immune cells than in other blood cells, and it is among the best nutrients modulating the immune system [148]. This is due to its antioxidant effect in polyunsaturated fatty acids (enhanced in membranes of immune cells), subject to oxidative damage because of their high metabolic activity and defense against pathogens [149,150]. In cancer, antitumor properties have been attributed to this group of compounds, especially γ- and δ-tocotrienols, because of their effect on molecular pathways involved in inhibition, apoptosis, and autophagy [151]. In our review (Appendix A), we analyzed two studies with tocotrienols and two with tocopherols. Both tocotrienol studies showed antitumor properties due to inhibition in growth, invasiveness, and migration [115,152]. Furthermore, plasma levels of tocopherols appeared to decrease with the progression of the disease [151]. In the cell culture model, supplementation with α- and γ-tocopherol showed VEGF inhibition [152]. Vit E compounds show antitumor properties in cell cultures such as the inhibition of proliferation, migration, and invasiveness and a decrease in apoptosis markers. However, the lack of studies in preclinical and clinical models does not allow us to conclude that these compounds are effective in breast cancer progression.

### 3.8. Iodine

Iodine is an essential micronutrient for the development of vertebrate organisms. It is a structural constituent of THs and a regulator of thyroid gland function [153]. Thyroid hormones play an essential role in the differentiation, growth, and energy metabolism of virtually all cells in the organism [154]. Furthermore, recent studies described that iodine, in its molecular form (I_2_), is a cellular modulator of organs capable of internalizing it, such as the breast, prostate, and pancreas, as well as the immune and nervous systems [155,156,157,158]. This chemical form of iodine has antioxidant [159], antineoplastic, and apoptotic effects in several cancer cells [160,161] and exhibits modulatory properties in the immune system [10]. Fresh seaweed is an important component of the Asian diet and is the only natural I_2_ source. Regular consumption of these algae is associated with a low incidence of breast diseases, such as fibrocystic disease or mastalgia and cancer, in these populations [156].

Various groups have shown the antineoplastic and immunomodulatory effects of I_2_ and proposed at least two mechanisms: (1) a direct action involving its antioxidant/oxidant properties and (2) an indirect effect through iodolipid formation. In the case of direct effects, two datasets were obtained showing that (a) at low or moderate concentrations, I_2_ significantly reduces lipid oxidation by competing with ROS for various cellular components or directly neutralizing HO radicals through coupling and generating iodinated species without oxidative activity i.e., hypoiodous acid (HOI) or hydrogen iodide (HI) [156,162], and (b) at high concentrations acting as a direct oxidant, I_2_ dissipates the mitochondrial membrane potential, inducing mitochondria-mediated apoptosis [163]. The indirect action involves the formation of iodolipids such as 6-iodo-5-hydroxy-8,11,14-eicosatrienoic acid (also called 6-iodolactone; 6-IL) derived from arachidonic acid (AA) iodination [164]. Concerning the mammary gland, it has been described that tumors induced by methyl nitrosourea (MNU) contain AA concentrations four times higher than normal tissue and that after chronic treatment (1 week) with oral I_2_ supplements, and 6-IL 15 times higher than in normal mammary tissue, suggesting that 6-IL plays a role in the antiproliferative effect of I_2_ [160]. These findings have also been corroborated in the human tumor cell line MCF-7 where lipids similar to 6-IL are detected after treatment with I_2_ [158] or apoptosis is triggered by I_2_ or 6-IL [165,166]. In this sense, our group described that the median lethal dose (LD_50_) of I_2_ for tumor cells is four times lower than that required for cells of normal origin, which suggests that the high availability of AA in tumor cells favors their iodination, generating 6-IL and triggering apoptosis [158].

Furthermore, we showed that 6-IL is a specific ligand and a potent promoter of peroxisome proliferator-activated receptor gamma (PPARγ) expression [167]. These receptors are ligand-activated transcription factors. In addition to regulating the expression of genes involved in lipid metabolism, their activation is associated with differentiation mechanisms, generating antiproliferative and drug resistance inhibition effects in various types of cancers [168]. Only three papers were yielded from the research carried out in PubMed (Appendix A). In preclinical models (MNU-induced mammary tumors in rats, in xenografts of various cancer cells in immunosuppressed mice, or canines with spontaneous breast cancer), the continuous oral supplement of I_2_ sensitized tumor cells, allowing a better antineoplastic response, decreasing tumor size, and avoiding chemoresistance [169,170,171]. In fact, in a murine model, the I_2_ supplement allowed reducing the doses of doxorubicin (DOX) up to fourfold, maintaining the antineoplastic effect and exerting protective effects on the heart and on health in general [169]. In a canine study, I_2_ supplementation, together with DOX neoadjuvant therapy, reduced the severity of side effects and improved tumor response. The tumor decline (18%) was accompanied by inhibition in the expression of resistance/invasion genes such as Survivin, drug resistance protein 1 (MDR1), and plasminogen activating urokinase (uPA). The 10-month survival analysis showed that I_2_ supplementation allowed a significant increase in disease-free time (73%) and survival (90%) [171]. In clinical studies in breast cancer patients, our group showed that the coadministration of I_2_ with FEC (5′-fluorouracil, epirubicin, cyclophosphamide) chemotherapy was accompanied by a greater antineoplastic response (25% decrease in tumor size) and the absence of chemoresistance processes observed in 30% of patients treated only with FEC. This effect correlated with the activation of Th1 antitumor immune signaling pathways and with overexpression of PPARγ receptors in FEC+I_2_ tumor samples. We also corroborated, as in the canine protocol, that the I_2_ supplement significantly attenuates intestinal, cardiac, and general health side effects [10].

In relation to the immune system’s modulatory mechanisms, it has been shown that various types of immune cells can internalize I_2_ and, depending on the cellular context, this element can act as an anti-inflammatory or proinflammatory agent. In vitro, I_2_ has also been shown to induce the release of antitumor cytokines, such as IL-6, IL-10, and IL-8 in normal leukocytes [172,173]. Another possibility currently explored in our laboratory is that I_2_ as an oxidized agent can exert epigenetic modifications associated with the activation of essential demethylase enzymes such as DNA methyltransferase 3 (DNMT3) ([174], unpublished data).

## 4. Discussion

Antineoplastic properties have been described in several micronutrients for decades, but none have shown solid evidence in vivo. One of the main pitfalls for any micronutrient is its bioavailability, which is usually low when supplementation is oral. For example, the average bioavailability is 33% for Vit D, 50% for Zn, 18% for iron, 15% for α-tocopherol, and 0.006% for Vit C [175,176,177]. All the micronutrients analyzed in this review have antiproliferative, apoptotic, and antimetastatic properties in vitro; however, in studies in vivo, the beneficial effects diminish or disappear. This can be explained because matching the dosages from in vitro to in vivo models orally and safely is difficult and often speculative. Another problem is the effectiveness; the heterogeneity of tumors and their differential response to treatments make it necessary to evaluate each nutrient for each type of malignancy. The third stumbling block is establishing the therapeutic dose/supplementation time. Not only have contrary effects been described depending on the timing of nutrient administration (as in the case of folates and Vit C), but numerous nutrients show different results depending on the dosage.

From the various mechanisms proposed to explain the antineoplastic effects of micronutrients, the most common is related to the antioxidant capability and includes Vit C and E, metals such as Zn, iron, and Se, and I_2_. In the case of Vit D, its effects are explained by the ability of its key molecule, calcitriol, to bind nuclear receptors and regulate gene expression. During tumor progression, folate treatments increase expression in DNA methylation enzymes (DNMT1), decreasing tumor suppression genes. Studies on I_2_ show that, in addition to its antioxidant actions in its 6-IL form, it is a genomic modulator as an agonist of PPARγ [160]. It has also been proposed as an epigenetic modifier due to its ability to regenerate DNA demethylating enzymes, which results in increased expression of tumor suppressor genes and genes of the cytotoxic immune system [178]. In this review, we analyzed the work of the main micronutrients in breast cancer progression. Only Vit D and I_2_ showed clear antitumor effects in clinical studies, and both nutrients possess the capacity for gene regulation. In their study, Madden et al. [9] administered chronically with low doses of Vit D (10 µg/day) and observed a 20% reduction in mortality (49%). The work of Moreno-Vega et al., where they showed the efficacy of I_2_ supplementation (alone or combined with chemotherapy) in a 5-year pilot study, showed a 63% increase in disease-free time, a reduction in tumor size, and cytotoxic immune system activation [10]. In this direction, there are many works analyzing the combination of nutrients and chemotherapeutic therapies evidencing synergic interactions which can lead to better outcomes [179,180]. Moreover, guidelines with combinations of different nutrients for cancer patients were commissioned by ESPEN (European Society for Clinical Nutrition and Metabolism) and by the European Partnership for Action Against Cancer (EPAAC) [181]. However, for now, more clinical studies are needed to establish their antitumor properties in vivo.

## 5. Patents

Aceves C, Anguiano B, Delgado G, Alfaro-Hernández Y, Torres-Martel JM, Peralta G, Domínguez A, Nava-Villalba M, Sosa S, Bontempo A, Godoy-García BL. Combinación de yodo molecular y antraciclinas de uso humano para la prevención y tratamiento de canceres quimiorresistentes captadores de yodo. Register: IMPI: MX/E/2017/009914. 19/04/2017. Validity 14/11/2012–14/11/20132.

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
