# Peer review of "Micronutrients and Breast Cancer Progression: A Systematic Review"

_nutrients, 2020, doi:10.3390/nu12123613_

Round 1
Reviewer 1 Report
This comprehensive systematic review summarizes the actual clinical effects of micronutrients in breast cancer. I have the following remarks:
- The dosage of micronutrients used in vitro is often higher than the dose that can safely be obtained in vivo, which (in part) explains the different results, and the authors should point this out in their Discussion.
- Another reason for failure of micronutrients in these patients, could be the absence of cancer-associated cachexia, being often less prevalent in breast cancer, and the authors should also note this in their Discussion.
- Some studies argue that combinations of these nutrients (e.g. ANICA trial), or even a complete nutritional approach (e.g. ESPEN guidelines) may be more effective than individual micronutrients, and the authors should point this out in their Discussion.
- Concerning vitamin E, it has even been shown that suppletion was detrimental in smokers having lung cancer, contradicting preclinical assays. The authors should make note of this in the Results, as breast cancer also tends to be associated with smoking, and may yield a similar risk.
Author Response
REVIEWER 1
Point 1: The dosage of micronutrients used in vitro is often higher than the dose that can safely be obtained in vivo, which (in part) explains the different results, and the authors should point this out in their Discussion.
The referee is right, one of the problems when scaling to more complex models (preclinical and clinical) is the lack of bioavailability due to the digestion process (in oral supplements) or its metabolisms/degradation. Likewise, trying to calculate doses from in vitro models with cell availability in vivo models is difficult and often speculative. We include a paragraph referring to these observations in the discussion.
Point 2: Another reason for failure of micronutrients in these patients, could be the absence of cancer-associated cachexia, being often less prevalent in breast cancer, and the authors should also note this in their Discussion.
Indeed, cachexia is undoubtedly a determining factor in the body's response to various treatments. However, in our search it did not turn out to be a relevant topic in the articles we examined. Therefore it was not included in our analysis.
Point 3: Some studies argue that combinations of these nutrients (e.g. ANICA trial), or even a complete nutritional approach (e.g. ESPEN guidelines) may be more effective than individual micronutrients, and the authors should point this out in their Discussion.
Yes, the referee is right, we found many articles with nutrient blends that were not eligible for this review because they did not analyze the nutrients separately to see their individual effects. In addition, there are many works that analyze the combination of nutrients and chemotherapeutic therapies that show synergistic interactions that can lead to better results. We include a paragraph referring to this aspect with 3 new references (182-184) in the discussion
Point 4: Concerning vitamin E, it has even been shown that suppletion was detrimental in smokers having lung cancer, contradicting preclinical assays. The authors should make note of this in the Results, as breast cancer also tends to be associated with smoking and may yield a similar risk.
Yes, smoking is indeed a risk factor for many cancers, including breast cancer, and its antagonism with some antioxidants such as vitamin C has been evidenced. We mentioned in the Vitamin C section.
Reviewer 2 Report
In this article Cuenca-Micó O. and Aceves C., focused on revising the effect of micronutrient intake or supplementation in breast cancer progression, which include Vitamins (C, D, and E), Folic Acid, Metals (Cu, Fe, Se, and Zn), Fatty acids, Polyphenols, and Iodine.
Overall, the scientific article is well-described. The tables are appropriate in presenting the results, both in an easy and understandable way. However, minor changes are suggested for publication acceptance:
- The abstract indicates the selected micronutrients to be analyzed and includes Vitamin A. However, in the Materials and Methods section (Page 2, lines 53-54) states that Vitamin A was less represented in the articles and will not be analyzed.
- In the results section (Page 2, lines 56-61), when considering the percentage of papers on each nutrients, this sums to 85%, where is the other 15%?
- A few sentences are missing references:
- Page 3, lines 136-138 - "The addition of Cu...and Act phosphorylation".
- Page 6, lines 245-253 - "Systemic Vit C...in other models".
- Some sections are referencing the Tables and other sections are not, all sections could reference the tables presented.
Author Response
REVIEWER 2
Point 1: The abstract indicates the selected micronutrients to be analyzed and includes Vitamin A. However, in the Materials and Methods section (Page 2, lines 53-54) states that Vitamin A was less represented in the articles and will not be analyzed.
The referee is right, we deleted Vitamin A in the Abstract.
Point 2: In the results section (Page 2, lines 56-61), when considering the percentage of papers on each nutrient, this sums to 85%, where is the other 15%?
The percentage of each nutrient was referred to the 425 results obtained from the first search, as many as 85 (20%) of papers in these 425 were nothing about any micronutrient.
Point 3:A few sentences are missing references:
Page 3, lines 136-138 - "The addition of Cu...and Act phosphorylation".
The reference for the page 3 comment is just at the end of the next sentence due both referred the same paper. we modify the paragraph to make it more explicit
The addition of Cu to breast cancer cells showed opposite results depending on the cell type. In triple-negative MDA-MB-231 cells with mutated p53, the Cu supplement increased proliferation and survival and Akt phosphorylation [56].
In contrast, in MCF-7 cells positive for ER, this micronutrient increased p53 phosphorylation, the expression of p21, resulting in cell cycle arrest in G1 and apoptosis [56].
Page 6, lines 245-253 - "Systemic Vit C...in other models”.
Page 6: All references about these concepts are above the text. We modify the paragraph
Systemic Vit C levels are inversely related to exposure to oxidants (such as tobacco). This decrease is also observed during disease progression, which could be explained, in part, by the increase in ROS activity characteristic of tumor progression [115,116]. These effects, such as the REdOx pair of Vit C, depend on its concentration. An antioxidant effect is observed at a physiological range of serum Vit C between 26-84 µM (equivalent to an intake of 75-90 mg/day). To achieve an oxidizing effect, at least> 100 µM in plasma is required, which is only achieved with intravenous injections of ascorbic acid [117].
Point 4: Some sections are referencing the Tables and other sections are not, all sections could reference the tables presented.
The referee is right, we acknowledge this as a mistake and will be corrected as suggested.